# PointCNT: A One-Stage Point Cloud Registration Approach Based on Complex Network Theory

Xin Wu [1], Xiaolong Wei [1,*], Haojun Xu [1], Caizhi Li [1], Yuanhan Hou [1], Yizhen Yin [2] and Weifeng He [1]

1   National Key Lab of Aerospace Power System Safety and Plasma Technology, Air Force Engineering University, Xi'an 710038, China; wuxin18892059516@163.com (X.W.); xuhaojun1965@163.com (H.X.); lcz_coco@163.com (C.L.); m13201659925@163.com (Y.H.); hehe_coco@163.com (W.H.)
2   School of Mechanical Engineering, Xi'an Jiaotong University, Xi'an 710049, China; yinyizhen@stu.xjtu.edu.cn
*   Correspondence: wei18892022001@163.com; Tel.: +86-1889-202-2001

**Abstract:** Inspired by the parallel visual pathway model of the human neural system, we propose an efficient and high-precision point cloud registration method based on complex network theory (PointCNT). A deep learning network (DNN) design method based on complex network theory is proposed, and a multipath feature extraction network, namely, Complex Kernel Point Convolution Neural Network (ComKP-CNN) for point clouds is designed based on the design method. Self-supervision is introduced to improve the feature extraction ability of the model. A feature embedding module is proposed to explicitly embed the transformation-variant coordinate information and transformation-invariant distance information into features. A feature fusion module is proposed to enable the source and template point clouds to perceive each other's nonlocal features. Finally, a Multilayer Perceptron (MLP) with prominent fitting characteristics is utilized to estimate the transformation matrix. The experimental results show that the Registration Recall (RR) of PointCNT on ModelNet40 dataset reached 96.4%, significantly surpassing one-stage methods such as Feature-Metric Registration (FMR) and approaching two-stage methods such as Geometric Transformer (GeoTransformer). The computation speed is faster than two-stage methods, and the registration run time is 0.15 s. In addition, ComKP-CNN is universal and can improve the registration accuracy of other point cloud registration methods.

**Keywords:** point cloud registration; deep learning; complex network theory; nonlocal features





## 1. Introduction

With the rapid development of high-precision sensors such as Light Detection and Ranging (LiDAR), point clouds have become the primary data format used to represent the three-dimensional (3D) world [1]. In recent years, the demand for high-quality point cloud data has increased with the rapid development of automatic driving, digital twins, intelligent robots, industrial product quality inspection and other fields. However, these sensors can only capture 3D scene information in a certain view and cannot capture complete 3D scene information. Point cloud registration is a task that aligns two or more different point clouds by estimating the relative transformation between them [2]. Therefore, point cloud registration plays a unique and critical role in computer vision tasks.

However, there are many challenges in point cloud registration. Unlike images, point clouds are unstructured with sparsity and disorder. Point clouds have considerable noise due to the inherent shortcomings of scanning sensors. In addition, the problems of partial overlap and large differences in the 3D features of the same object from different views also bring challenges to point cloud registration.

Most traditional algorithms divide registration into two steps: first, find the correspondences, and then, estimate the rigid transformation matrix according to the correspondences. Obtaining the transformation matrix is simple when the correspondences

are known. Similarly, finding the correspondences when the transformation matrix is known is simple. Given these two observations, most algorithms alternate between these two steps to obtain a better result [3]. However, traditional algorithms usually have high computational complexity, require many iterations, take a long time to compute, do not meet real-time requirements, and are nonconvex. These considerations make them sensitive to the initial position and easily fall into the local optimal solution. The method proposed in this paper does not need to calculate correspondences or need iterations, so it has good real-time performance.

Deep learning has shown great advantages due to its prominent fitting characteristics and has been widely used in automatic driving, healthcare, machine translation, damage detection and other fields [4]. Robust and Efficient Point Cloud Registration using PointNet (PointNetLK) [5] pioneered the application of deep learning to point cloud registration. The application of deep learning in point cloud registration has made great progress and significantly improved registration robustness and efficiency. Unfortunately, most deep-learning-based methods do not deviate from the traditional algorithm design. Traditional design is divided into two steps: finding correspondences (or computing a soft matching matrix) and estimating the transformation matrix. These are called "two-stage methods" in this paper. This kind of method completely separates the module for finding correspondences from the module for estimating the transformation matrix. These modules are trained separately, which causes accumulative errors. Two-stage methods have high computing costs and poor real-time performance because they need to find correspondences and calculate their confidence to eliminate outliers. An end-to-end "one-stage method" with fast computation speed and good real-time performance is proposed in this paper. Instead of finding correspondences between the source point cloud and template point cloud, a deep learning model is used to directly extract the global features of the two point clouds, and then the transformation matrix is directly estimated according to the global features. This process makes our registration method robust to noise and able to handle the partial overlap problem.

At present, there are few studies on one-stage methods. Most studies apply only the deep learning model used for two-dimensional (2D) images to point cloud registration after simple changes. Although their registration accuracy exceeds most traditional algorithms, there is still a large gap compared with the two-stage methods. According to the point cloud data characteristics, a new one-stage framework is designed to improve registration accuracy.

In this work, a new efficient and high-precision one-stage point cloud registration method based on complex network theory (PointCNT) is designed, which estimates the rigid transformation matrix by global features without searching for correspondences. An overview of PointCNT is shown in Figure 1. Our method consists of four parts. (1) Feature extraction module. Inspired by the parallel visual pathways model [6], a multipath feature extraction network for point clouds based on complex network theory is designed. A self-supervised module is introduced to improve the feature extraction ability. (2) Feature embedding module. Inspired by nonlocal neural networks [7,8], Geometric-based Self-attention (GBSelf-attention) is designed. GBSelf-attention embeds the transformation-variant coordinate information and the transformation-invariant distance information with geometric consistency between points into the point cloud feature. (3) Feature fusion module. Feature-based Cross-attention (FBCross-attention) is designed to fuse the source and template features so that the extracted features of the two point clouds can be transmitted interactively. (4) Registration module. Multilayer Perceptron (MLP) [9–12] is used to estimate the rotation and translation, and the rotation quaternion is used to represent the point cloud rotation.

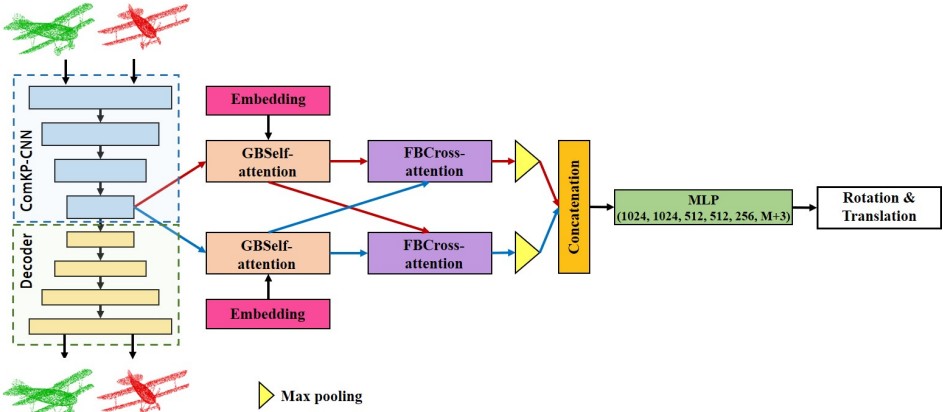

**Figure 1.** Architecture of the proposed network PointCNT.

To summarize, our main contributions are threefold:

(1) An efficient, high-precision and end-to-end one-stage point cloud registration framework is proposed.

(2) A deep learning network design method based on complex network theory is proposed, and a multipath feature extraction network based on the above method for point clouds is designed.

(3) A self-supervised module is introduced to improve the feature extraction ability of the network.

(4) GBSelf-attention and FBCorss-attention based on nonlocal neural networks are designed.

In Section 2, we summarize related research work on point cloud registration, mainly including traditional registration methods, learning-based two-stage methods and learning-based one-stage methods. In the Section 3, we introduce the cloud registration method PointCNT designed in this paper, which mainly includes feature extraction module, feature embedding module, feature fusion module and registration model. In Section 4, we carry out experiments to verify the effectiveness of PointCNT. We carry out ablation experiments to study the effects of different modules on the model and verify the performance of the designed feature extraction module, feature embedding module and feature fusion module. In Section 5, we discuss the research results. Section 6 is the conclusion.

## 2. Related Work

### 2.1. Traditional Registration Methods

Point cloud registration is divided into coarse registration and fine registration. Typical coarse registration algorithms include the Point Feature Histogram (PFH) [13], Fast Point Feature Histogram (FPFH) [14], 3D Shape Context (3Dsc) [15], Normal Distributions Transform (NDT) [16], 4-Points Congruent Sets (4PCS) [17] and Principal Component Analysis (PCA) [18]. The coarse registration algorithm is not sensitive to the initial pose, but its registration accuracy is low. The coarse registration can be considered a preprocessing process for point cloud initialization in fine registration. Iterative Closest Point (ICP) and its variants [19–22] are the best-known traditional fine registration algorithms. ICP alternates between finding point cloud correspondences and solving a least-squares problem to update the alignment. However, ICP-style methods are prone to local minima due to nonconvexity. To solve the above problem, a Globally Optimal Solution to 3D ICP Point-set Registration (Go-ICP) [23] uses a branch-and-bound method to search the motion space. Go-ICP outperforms local ICP methods when a global solution is desired but is several orders of magnitude slower than other ICP variants. Traditional methods do not require a large quantity of training data and have excellent generalization ability. However, they are usually sensitive to noise, have difficulty processing partially overlapping point clouds, easily converge to local optimal solutions, have low registration accuracy and have long computational times. Unlike traditional registration methods, PointCNT based on deep

learning is an end-to-end algorithm. It is insensitive to noise and can process partial overlap problems with high computational efficiency and registration accuracy.

### 2.2. Learning-Based Two-Stage Registration Methods

At present, most research adopts two-stage methods to estimate the transformation matrix, as shown in Figure 2 and Tabel 1. In the first stage, the correspondences between the source and template are predicted, such as the corresponding relationship of key points, the corresponding relationship of feature points and the corresponding relationship of all points. In the second stage, the transformation matrix is estimated according to the correspondences. In this stage, Singular Value Decomposition (SVD) [19], Random Sample Consensus (RANSAC) [24] or Artificial Neural Networks (ANNs) [25] are usually used to estimate the transformation matrix.

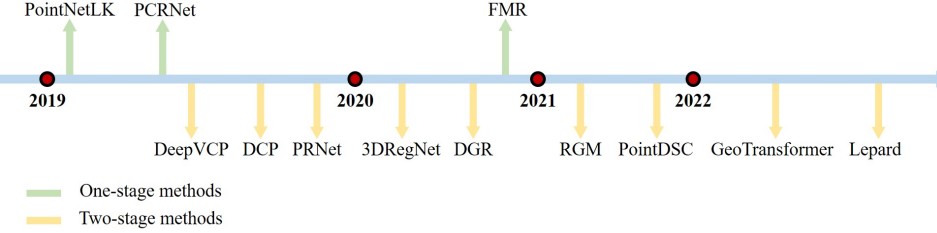

**Figure 2.** The development of point cloud registration methods based on deep learning.

**Table 1.** Comparison of different point cloud registration methods.

| Method | Category | Proposed Year | Advantage | Disadvantage |
|---|---|---|---|---|
| ICP | | 1992 | No need for a large amount of data for training. | Sensitive to the initial position of the point cloud and prone to falling into local optima. |
| Go-ICP | Traditional Registration Method | 2016 | Adopting a global solution for higher registration accuracy. | Running speed is very slow. |
| DCP | | 2019 | Has good robustness to noise. | Not applicable for partial overlap. |
| PointDSC | | 2021 | High registration accuracy, suitable for partial overlap. | Slow running speed, requires additional algorithms to find corresponding points. |
| GeoTransformer | Learning-based Two-stage Method | 2022 | High registration accuracy, suitable for partial overlap, without the need for additional algorithms to find corresponding points. | Slow running speed, registration accuracy constrained by key point matching. |
| PointNetLK | | 2019 | Applying deep learning to point cloud registration for the first time. | Low registration accuracy, robustness and poor generalization. |
| PCRNet | | 2019 | Has good robustness to noise, is an end-to-end model, and runs fast. | The model has a simple structure and low registration accuracy. |
| FMR | Learning-based One-stage Method | 2020 | The unsupervised learning method is used to extract point cloud features, and the inverse synthesis algorithm is used to calculate the transformation matrix. | Poor registration performance when applied to point clouds with only partial overlap. |

As a classical two-stage method, Deep Closest Point (DCP) [3] first extracts the local point features in a point cloud, then establishes the soft matching matrix among points based on the extracted features and then uses weighted SVD to compute the transformation matrix according to the soft matching matrix. DCP is robust to noise, but its performance is poor when applied to point clouds with only partial overlap. Deep Global Registration (DGR) [26] is similar to DCP, but DGR changes the gradient propagation mode of weighted SVD. DGR takes the derivative of the loss function with respect to the weight $w$, reducing the computational complexity and improving the registration accuracy. Deep

Virtual Corresponding Points (DeepVCP) [2], Partial Registration Network (PRNet) [27] and Geometric Transformer (GeoTransformer) [28] all use key points for matching. First, point cloud features are extracted by DNN, and key points are obtained according to the features. Then, the correspondence matrix is established according to the key points, and the transformation matrix is estimated according to the correspondence matrix. This kind of algorithm further improves the registration accuracy and can handle the partial overlap problem. Deep Neural Network for 3D Point Registration (3DRegNet) [29] and Robust Point Cloud Registration using Deep Spatial Consistency (PointDSC) [7] directly take correspondences as input, use ANN to eliminate the outliers, and then estimate the transformation matrix according to the correspondences that eliminated outliers. This kind of algorithm focuses on the outlier elimination method and obtains the correspondences with a higher proportion of inliers to improve the registration accuracy. Robust Point Cloud Registration Framework Based on Deep Graph Matching (RGM) [30] introduces the idea of a graph, such that the point features not only include the local geometric information but also include the structure and topology information in a wider range to find more correct correspondences.

Two-stage methods usually combine SVD to obtain the registration transformation matrix. Their accuracy is high, but they need to find the correspondences and compute the confidence to eliminate the outliers. Therefore, the computational cost is high, and the real-time performance is poor. Compared with the two-stage methods, our method does not need to find the corresponding point relationships. Our method directly estimates the transformation matrix according to the global features. This process avoids the accumulative errors caused by the complete separation of the feature extraction network and the module for computing the transformation matrix and improves the computational speed.

*2.3. Learning-Based One-Stage Registration Methods*

PointNetLK [5] pioneered the application of deep learning to point cloud registration. First, MLP and max pooling are used to extract the global features, then the inverse synthesis algorithm is used to improve the Lucas–Kanade Algorithm (LK) [31] and the improved LK algorithm is used to estimate the transformation matrix according to the global features. Point Cloud Registration Network using PointNet Encoding (PCRNet) [32] first utilizes MLP and max pooling to extract the global features of the source and template, concatenates the two global features, and then inputs the features into the ANN to estimate the transformation matrix. PCRNet is robust to noise because the transformation matrix is computed based on the global features. However, its structure is simple, and the registration accuracy is lower than that of the two-stage model. Feature-Metric Registration (FMR) [33] utilizes a self-supervised learning model composed of an encoder and decoder to extract features. Then, the transformation matrix is computed by the inverse synthesis algorithm based on the extracted features. However, this method performs poorly when dealing with partial overlap.

The one-stage method does not need to find the correspondences between the source and template but directly computes the transformation matrix according to the extracted features, which has a fast computation speed and good real-time performance. However, at present, there is little research on the one-stage method. The model structure is simple, and the registration accuracy of the one-stage model is still lower than that of the two-stage model. This paper proposes a new and complex one-stage method framework for registration. Inspired by the parallel visual pathway model in the human neural system, a novel feature extraction network is designed based on complex network theory. Additionally, a self-supervised model is introduced to improve the feature extraction ability of the network. GBSelf-attention and FBCorss-attention are designed to integrate the source point cloud and template point cloud features. The registration accuracy of our method is significantly higher than that of the above one-stage method and achieves state-of-the-art performance. PointCNT is also robust to noise, suitable for partial overlap and has a high inference speed.

## 3. PointCNT

Given two point clouds $Q = \{q_i \in \mathbb{R}^3 \mid i = 1, 2, \cdots, N\}$ and $P = \{p_t \in \mathbb{R}^3 \mid i = 1, 2, \cdots, M\}$, the point cloud registration goal is to estimate a rigid transformation $\mathbf{T} = \{\mathbf{R} \in SO(3), \mathbf{t} \in \mathbb{R}^3\}$ that aligns the two point clouds with a rotation matrix $\mathbf{R}$ and a translation vector $\mathbf{t}$. The transformation can be solved by

$$\hat{\mathbf{R}}, \hat{\mathbf{t}} = \underset{\mathbf{R} \in SO(3), t \in \mathbb{R}^3}{\arg \min} \sum_{(p_{x_i}, q_{y_i}) \in C} \rho\left(q_{y_i}, \mathbf{R}p_{x_i} + \mathbf{t}\right), \tag{1}$$

where $C$ is the set of ground-truth correspondences between $Q$ and $P$, and $\rho(a, b)$ is some distance. However, in this paper, $C$ is not solved, and $\mathbf{T} = \{\mathbf{R} \in SO(3), \mathbf{t} \in \mathbb{R}^3\}$ is directly solved according to the global features.

The pipeline of our network PointCNT is shown in Figure 1 and can be summarized as follows:

$$\hat{\mathbf{R}}, \hat{\mathbf{t}} = R\{\varphi[\mathrm{E}(\phi(P)), \mathrm{E}(\phi(Q))]\}, \tag{2}$$

where $\phi(\bullet)$ is the feature extraction module, $\mathrm{E}(\bullet)$ is the feature embedding module, $\varphi[\bullet]$ is the feature fusion module and $R[\bullet]$ is the feature registration module.

In this section, we introduce the one-stage point cloud registration method PointCNT in detail. In Section 3.1, we design the point cloud feature multipath extraction network ComKP-CNN based on the complex network theory, and introduced the self-monitoring method to enhance the ability of the network to extract point cloud features. In Section 3.2, we design GBSelf-attention to explicitly embed the coordinate information and distance information of the point cloud into the features. In Section 3.3, we design FBCross-attention to realize the interactive propagation of features between the source and template. In Section 3.4, we realize the registration of point clouds through MLP. Section 3.5 describes the loss function used in this paper.

### 3.1. Feature Extraction Module

In this section, Kernel Point Convolution (KPConv) [34] is used as the basic module for extracting point cloud features. Inspired by the parallel visual pathway model in the human neural system, as shown in Figure 3, a multipath feature extraction network based on complex network theory is designed. The parallel visual pathways model considers that the high-level brain regions related to vision do not simply receive signals from the retina through one neural pathway but receive neural signals through multiple pathways, and the number of neurons between different pathways is different. The network for extracting point cloud features should have a similar topology structure to the parallel visual pathways model. It is a complex network in which features have multiple transmission pathways rather than a single pathway.

A large number of empirical studies [35–37] show that networks in the real world are complex networks between regular networks and random networks, as shown in Figure 4. Almost all of these networks have a small-world effect; that is, networks have a smaller average path length, as shown in Equation (3), and a larger clustering coefficient, as shown in Equation (4). However, at present, most DNNs used to extract point cloud features [34,38–40] or even used to extract 2D image features [41–44] are not complex networks but regular networks, which do not have a small-world effect, as shown in Table 2. Based on the above analysis, we propose a new DNN design method based on complex network theory and use this method to design a new point cloud feature extraction network.

$$L = \frac{1}{\frac{1}{2}N(N-1)} \sum_{i \geq j} d_{ij}, \tag{3}$$

where $N$ is the number of network nodes, and $d_{ij}$ is the path length between node $i$ and node $j$.

$$C = \frac{1}{N} \sum_{i=1}^{N} \frac{2R_i}{k_i(k_i - 1)}, \qquad (4)$$

where $N$ is the number of network nodes, $R_i$ is the number of triangles formed by node $i$ and its neighbor nodes and $k_i$ is the number of first-order neighbor nodes of node $i$.

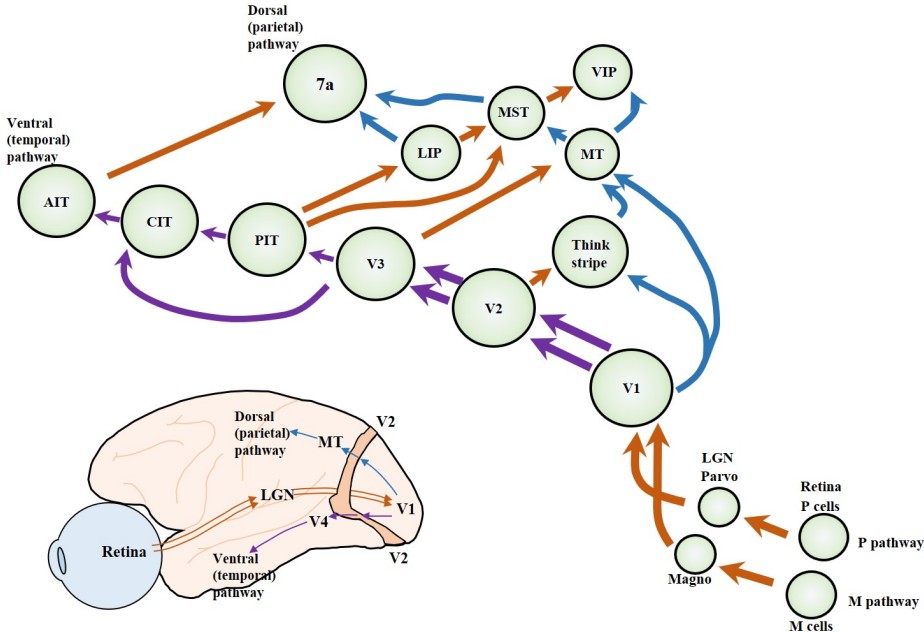

**Figure 3.** Parallel visual pathways model.

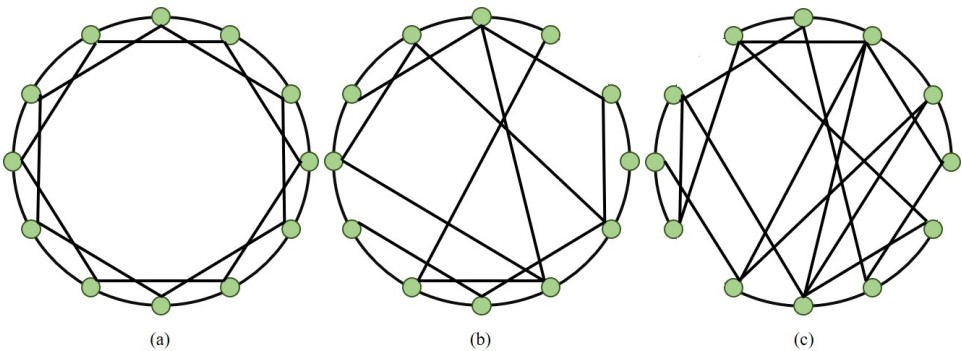

**Figure 4.** (**a**) Regular network. (**b**) Complex network. (**c**) Random network.

**Table 2.** The average path length and clustering coefficient of a typical DNN.

|  | DNN | Average Path Length | Clustering Coefficient |
|---|---|---|---|
| DNN for images | VGG16 | 5.647 | 0 |
|  | ResNet50 | 6.93 | 0 |
| DNN for point clouds | KP-CNN | 3.972 | 0 |
|  | ComKP-CNN | 1.597 | 0.684 |

The DNN design method proposed in this paper includes three steps, as shown in Figure 5. First, the existing network designed by researchers is selected as the backbone, and the network is extended to a global coupling network. Then, the network is trained to obtain each edge weight. If the edge weight is small, the edge is considered to play a small role in extracting features, so the edge is deleted. Then, the network is retrained to obtain a

complex network with a small-world effect. Based on complex network theory, this method can design a DNN with excellent performance and can better extract the input data features. This design can be used not only in the design of point cloud feature extraction networks but also in the design of feature extraction networks for images, text, voice and other types of data.

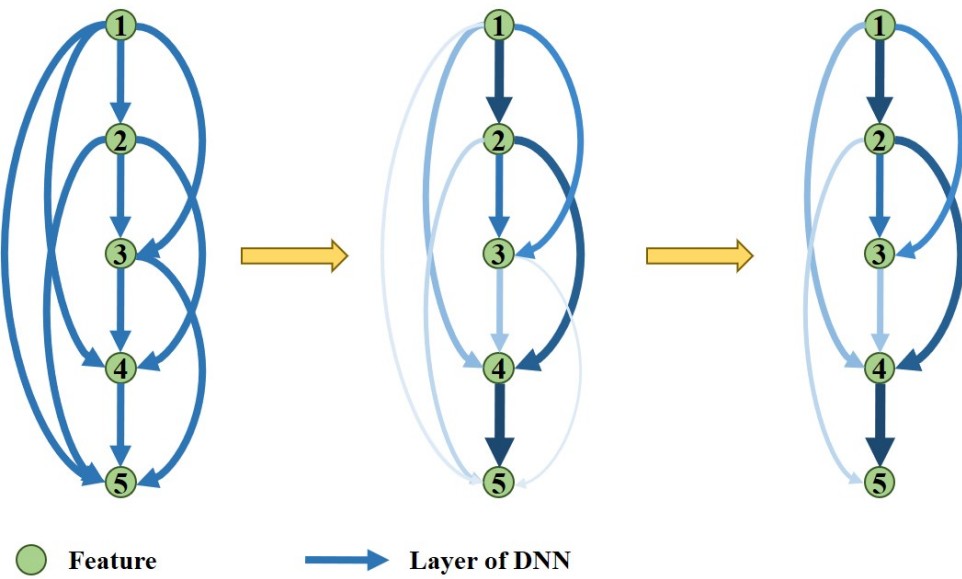

**Figure 5.** The proposed DNN design method is based on complex network theory.

In this paper, a Kernel Point Convolution Neural Network (KP-CNN) [34] is used as the backbone to sample the point cloud. To prevent gradient explosion and gradient disappearance, the residual structure in KP-CNN is retained. The feature extraction module is designed using the above network design method, called the Complex Kernel Point Convolution Neural Network (ComKP-CNN). The KP-CNN used in this paper includes a KPConv Block (ConvBlock), as shown in Equation (5), and 10 Residual Blocks (ResBlock), as shown in Equation (6).

$$F_{out} = AF\{GN[\Theta(F_{in})]\}, \tag{5}$$

where $F_{in}$ is the input features, $F_{out}$ is the output features, $\Theta(\bullet)$ is KPConv, $GN[\bullet]$ is group normalization, and $AF\{\bullet\}$ is the activation function. LeakyReLU is adopted in this paper.

$$F_{out} = UB_2\{CB[UB_1(F_{in})]\} + UB_3[Max(F_{in})], \tag{6}$$

where $CB[\bullet] = AF\{GN[\Theta(\bullet)]\}$ is ConvBlock, $UB(\bullet) = AF\{GN[MLP(\bullet)]\}$ is a unary block, which is mainly responsible for integrating the feature channels, $MLP(\bullet)$ is MLP and $Max(\bullet)$ is max pooling. When the channels of $F_{in}$ are not equal to the channels of $F_{out}$, $UB_3(F) = AF\{GN[MLP(F)]\}$; otherwise, $UB_3(F) = F$. When the point cloud is sampled down by the ResBlock, $Max(\bullet)$ is used to sample the input features at the short edge; otherwise, $Max(\bullet)$ is not used.

KP-CNN constructs a simple chain network with features as nodes and feature extraction layers as edges. In this paper, $UB_2\{CB[UB_1(F_{in})]\}$ (AddBlock) is used as the added edge to build a global coupling network, which is called the global coupling KP-CNN, as shown in Figure 6a. Then, we train the network, remove the edges with small weights and retrain the network to obtain a complex network with a small-world effect, called ComKP-CNN, as shown in Figure 6c.

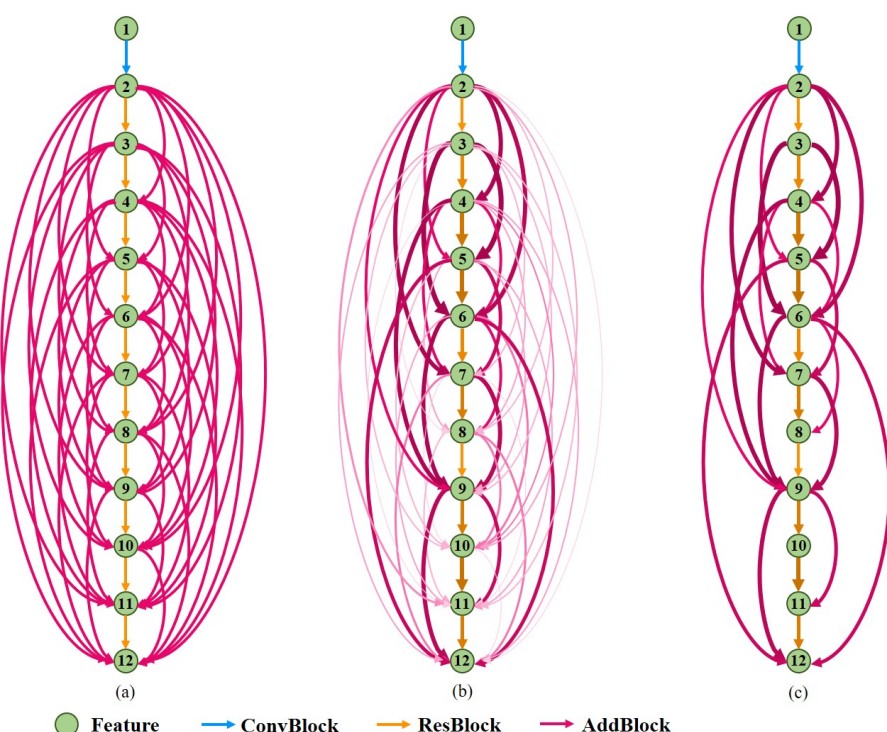

**Figure 6.** The structure of the global coupling KP-CNN (**a**), global coupling KP-CNN after training (**b**) and ComKP-CNN (**c**).

The feature extracted from ComKP-CNN is input to the decoder, and the coordinates of each point are output. The KP-FCNN is utilized as the decoder, which consists of nearest upsampling and unary convolution. Features are transmitted from the intermediate layers of the encoder to the decoder through skip links, as shown in Figure 7.

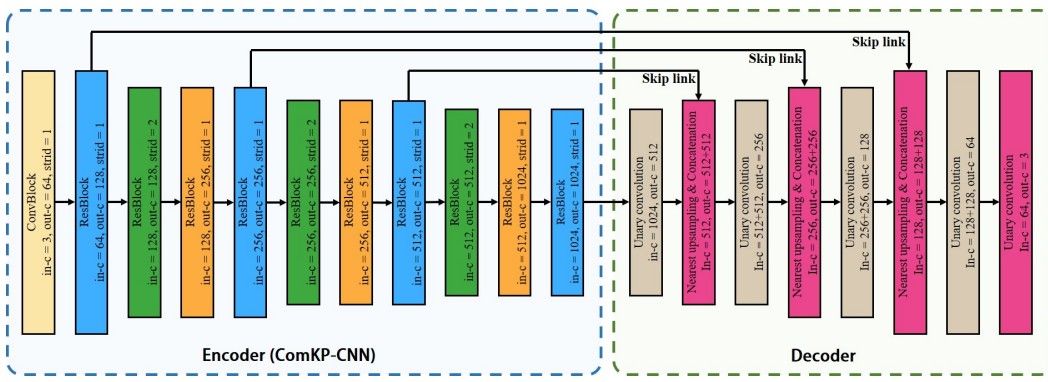

**Figure 7.** The structure of the self-supervised model.

## 3.2. Feature Embedding Module

### 3.2.1. GBSelf-Attention

Global context has proven critical in many computer vision tasks [28,45,46]. Since our model estimates the transformation matrix through the global features of the point cloud, rather than through the correspondences, the model needs to obtain the transformation-variant information of the point cloud. Therefore, the point coordinates are explicitly embedded into the features so that the features have transformation-variant characteristics. Additionally, the geometric features of the overlapping part of the source point cloud and template point cloud have geometric consistency, so we explicitly embed the distance information between points with transformation invariance into the features. Inspired by nonlocal neural networks, we design a geometric GBSelf-attention, as shown in Figure 8, to

learn the global correlations in both feature and geometric spaces among the downsampled points for each point cloud. We describe the computation for downsampled points $\tilde{P}$, and the same goes for $\tilde{Q}$. The feature of $\mathbf{F}_{\tilde{P}} \in \mathbb{R}^{|\tilde{P}| \times d_f}$ is taken as the input feature of GBSelf-attention and the output feature $\tilde{P}$, and the same goes for $\tilde{Q}$. The feature of $\mathbf{F}_{\tilde{P}GB} \in \mathbb{R}^{|\tilde{P}| \times d_f}$ can be computed by

$$\mathbf{F}_{\tilde{P}GB} = MLP[softmax(\mathbf{S}_{\tilde{P}\tilde{P}})(\mathbf{F}_{\tilde{P}}\mathbf{V})] + \mathbf{F}_{\tilde{P}}, \tag{7}$$

where $MLP[\bullet]$ is the MLP, $softmax(\bullet)$ is a row-wise softmax function and $\mathbf{S}_{\tilde{P}\tilde{P}} \in \mathbb{R}^{|\tilde{P}| \times |\tilde{P}|}$ is the attention score matrix, which can be computed as

$$\mathbf{S}_{\tilde{P}\tilde{P}} = \frac{\sum\limits_{i=1}^{|\tilde{P}|} \left[ (\mathbf{F}_{\tilde{P}}\mathbf{Q})(\mathbf{E}_i\mathbf{R})^T \right] + (\mathbf{F}_{\tilde{P}}\mathbf{Q})(\mathbf{F}_{\tilde{P}}\mathbf{K})^T}{\sqrt{d_f}}, \tag{8}$$

where $\mathbf{Q}, \mathbf{K}, \mathbf{V}, \mathbf{R} \in \mathbb{R}^{d_f \times d_f}$ are the respective projection matrices for queries, keys and values, $\mathbf{E} \in \mathbb{R}^{|\tilde{P}| \times |\tilde{P}| \times d_f}$ is the geometric structure embedding and $\mathbf{E}_i \in \mathbb{R}^{|\tilde{P}| \times d_f}$ is the $i$th element of $\mathbf{E}$.

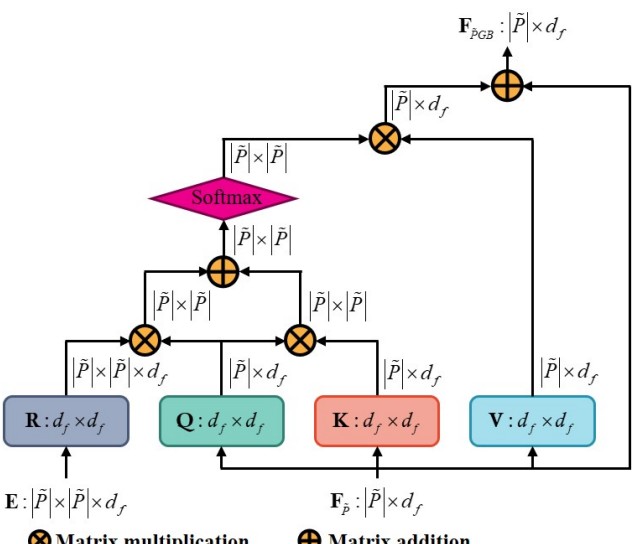

**Figure 8.** The computation graph of GBSelf-attention.

### 3.2.2. Coordinate Embedding

The coordinate embedding $e_{i,j}^C$ between $\tilde{p}_i$ and $\tilde{p}_j$ is computed by Equation (9):

$$e_{i,j}^C = \frac{1}{2} \frac{x_i + y_i + z_i + x_j + y_j + z_j}{|x|_{\max} + |y|_{\max} + |z|_{\max}}, \tag{9}$$

where $(x_i, y_i, z_i)$ and $(x_j, y_j, z_j)$ are the coordinates of $\tilde{p}_i$ and $\tilde{p}_j$, respectively, and $|x|_{\max}$, $|y|_{\max}$ and $|z|_{\max}$ are the maximum distances between point cloud $\tilde{P}$ and the origin along the coordinate axis.

### 3.2.3. Distance Embedding

Give any two points $\tilde{p}_i, \tilde{p}_j \in \mathbb{R}^3$ in $\tilde{P}$, and define the distance between them as $d_{i,j} = \left\| \tilde{p}_i - \tilde{p}_j \right\|_2$. The distance embedding $e_{i,j}^D$ between them is computed by applying a sinusoidal function [47] on $d_{i,j}/\alpha_d$. Here, $\alpha_d$ is a hyperparameter used to tune the sensitivity to distance variations.

Finally, the geometric structure embedding $\mathbf{e}_{i,j}$ is computed by aggregating the coordinate embedding and the distance embedding:

$$\mathbf{e}_{i,j} = copy\left(e_{i,j}^{C}, d_f\right)\mathbf{C} + copy\left(e_{i,j}^{D}, d_f\right)\mathbf{D}, \tag{10}$$

where $copy(x, d) \in \mathbb{R}^d$ represents copying $x$ as a vector with dimension $d$, and $\mathbf{C}, \mathbf{D} \in \mathbb{R}^{d_f \times d_f}$ are the respective projection matrices for the distance embedding and the coordinate embedding.

### 3.3. Feature Fusion Module

Given the GBSelf-attention feature $\mathbf{F}_{\tilde{P}GB}$, $\mathbf{F}_{\tilde{Q}GB}$ with the distance and coordinate embedding for $\tilde{P}$ and $\tilde{Q}$, respectively, the FBSelf-attention feature $\mathbf{F}_{\tilde{P}FB}$ of $\tilde{P}$ is computed with the GBSelf-attention feature $\mathbf{F}_{\tilde{P}GB}$ and $\mathbf{F}_{\tilde{Q}GB}$:

$$\mathbf{F}_{\tilde{P}FB} = MLP\left[softmax\left(\mathbf{S}_{\tilde{P}\tilde{Q}}\right)\left(\mathbf{F}_{\tilde{Q}GB}\mathbf{V}\right)\right] + \mathbf{F}_{\tilde{P}GB}, \tag{11}$$

where $\mathbf{S}_{\tilde{P}\tilde{Q}} \in \mathbb{R}^{|\tilde{P}| \times |\tilde{Q}|}$ is the attention score matrix, which is computed as the feature correlation between $\mathbf{F}_{\tilde{P}GB}$ and $\mathbf{F}_{\tilde{Q}GB}$:

$$\mathbf{S}_{\tilde{P}\tilde{Q}} = \frac{(\mathbf{F}_{\tilde{P}GB}\mathbf{Q})\left(\mathbf{F}_{\tilde{Q}GB}\mathbf{K}\right)^{T}}{\sqrt{d_f}}. \tag{12}$$

GBSelf-attention embeds the coordinate information as transformation-variant and the distance information as transformation-invariant into each individual point cloud so that the features can explicitly capture the geometric structure information. FBCors-attention enables two point clouds to perceive each other's features so that the geometric consistency of the overlapping part can be transmitted interactively between the two point clouds. Finally, symmetric function max pooling is used to capture the global features $F_{\tilde{P}g} \in \mathbb{R}^{d_f}$ and $F_{\tilde{Q}g} \in \mathbb{R}^{d_f}$ and stacks $F_{\tilde{P}g}$ and $F_{\tilde{Q}g}$ in the channel dimension to obtain $F_{\tilde{P}\tilde{Q}} \in \mathbb{R}^{2d_f}$. The process is as follows:

$$F_{\tilde{P}\tilde{Q}} = Cat\left[Max(F_{\tilde{P}FB}), Max\left(F_{\tilde{Q}FB}\right)\right], \tag{13}$$

where $Cat[a, b]$ represents stacking $a$ and $b$ in the channel dimension, and $Max(\bullet) : \mathbb{R}^{|\tilde{P} \text{ or } \tilde{Q}| \times d_f} \to \mathbb{R}^{d_f}$ represents the max pooling of point cloud features in the dimension of points.

### 3.4. Registration Module

MLP is used to estimate the transformation matrix because of its prominent fitting characteristics. The registration module has five hidden layers, 1024, 1024, 512, 512, 256, and an output layer of size $M + 3$, whose parameters represent the estimated transformation $\mathbf{T}$. The first $M$ of the output values are used to represent the rotation, and last three represent the translation vector $\hat{\mathbf{t}} \in \mathbb{R}^3$. The rotation matrix $\hat{\mathbf{R}} \in SO(3)$ can represent the point cloud rotation, where $M = 9$, or by the rotation quaternion $\hat{\mathbf{q}} \in so(3)$, where $M = 4$. The experimental results show that PointCNT achieves better registration results when the rotation quaternion is used to represent rotation. Therefore, we use the rotation quaternion to represent point cloud rotation.

### 3.5. Loss Function

The loss function $L$ consists of registration loss $L_{Reg}$ and self-supervised loss $L_{Unsup}$:

$$L = L_{Reg} + \alpha L_{Unsup}, \tag{14}$$

where $\alpha \in (0, 1)$ is the self-supervised coefficient, which is used to balance the role of the self-supervised module on the model.

Referring to DCP [3], we use the following loss function to measure our model's agreement with the ground-truth rigid motions:

$$L_{Reg} = \left\| \hat{\mathbf{R}}^T \mathbf{R}_g - \mathbf{I} \right\|_2^2 - \left\| \hat{\mathbf{t}} - \mathbf{t}_g \right\|_2^2 - \lambda \|\theta\|_2^2, \tag{15}$$

where $\hat{\mathbf{R}}$ and $\hat{\mathbf{t}}$ represent the rotation matrix and translation vector estimated by PointCNT, respectively, and $\mathbf{R}_g$ and $\mathbf{t}_g$ denote the ground truth. The first two terms define a simple distance on $SE(3)$. The third term denotes Tikhonov regularization of the PointCNT parameters $\theta$, which serves to reduce the network complexity.

A self-supervised module is introduced to enhance the feature extraction capability of our method. The loss function of the self-supervised module is as follows:

$$L_{Unsup} = \frac{1}{|P|} \sum_{i=1}^{|P|} \rho\big(p_i, \psi[\phi(F_{p_i})]\big) + \frac{1}{|Q|} \sum_{i=1}^{|Q|} \rho\big(q_i, \psi[\phi(F_{q_i})]\big), \tag{16}$$

where $\phi(\bullet)$ is the ComKP-CNN, $\psi[\bullet]$ is the decoder and $\rho(a, b)$ represents some distance between $a$ and $b$. In this paper, $\rho(a, b) = \|a - b\|_2$.

## 4. Experiments and Results

In this section, we carry out experiments to study the point cloud registration method proposed in this paper. In Section 4.1, we introduce the details of the experiment, including the dataset and evaluation metrics. In Section 4.2, our method is evaluated on the CAD simulation dataset ModelNet40 and the outdoor dataset KITTI. In Section 4.3, ablation experiments are carried out to study the effects of ComKP-CNN, self-supervised module, coordinate embedding, distance embedding, FBCross-attention, max pooling as symmetric function and rotation quaternion as the representation of point cloud rotation on the model. The improvement of ComKP-CNN on other point cloud registration methods is also studied, which verifies the performance of the DNN design method proposed in this paper.

### 4.1. Implementation Details

We implement PointCNT in PyTorch. The experiment was carried out on a single Graphic Processing Unit (GPU) server. The GPU is an NVIDIA GeForce RTX3090, and the operating system is Ubuntu 20.04. The initial learning rate is set to $10^{-4}$, and the Adam [48] optimization method and cosine annealing warm restart [49] learning rate adjustment method are utilized. All models are trained for 100 epochs.

#### 4.1.1. Dataset Used in the Experiments

ModelNet40 [50] contains 3D CAD models from 40 categories. It is a widely used dataset for training 3D deep learning networks. We split ModelNet40 into two parts, each of which contains 20 point cloud categories. One part is split into a training set and a testing set according to the proportion of 8:2 to perform same-category testing. The other part is used to perform cross-category testing. ModelNet40 is a simulation dataset with similar characteristics to industrial products. This paper conducts experiments on the ModelNet40 dataset because point cloud registration has been applied to industrial product quality inspection. To verify the effectiveness of our model, we also conduct experiments on the 3DMatch dataset and KITTI dataset. 3DMatch [51] contains 62 scenes, among which 46 are used for training, 8 for validation and 8 for testing. KITTI contains point clouds captured with a Velodyne HDL64 LiDAR in Karlsruhe, Germany, together with the "ground truth" poses provided by a high-end GNSS/INS integrated navigation system. KITTI [52] contains point clouds captured with a Velodyne HDL64 LiDAR in Karlsruhe, Germany, together

with the "ground truth" poses provided by a high-end GNSS/INS integrated navigation system.

### 4.1.2. Evaluation Metrics

We evaluate PointCNT with three metrics: (1) Relative Rotation Error (RRE), the geodesic distance between estimated and ground-truth rotation matrices; (2) Relative Translation Error (RTE), the Euclidean distance between estimated and ground-truth translation vectors; and (3) Registration Recall (RR), the fraction of point cloud pairs whose RRE and RTE are both below certain thresholds (i.e., RRE < 5° and RTE < 0.01).

$$RRE(\hat{\mathbf{R}}) = \arccos \frac{trace(\hat{\mathbf{R}}^T \mathbf{R}_g) - 1}{2}. \tag{17}$$

$$RTE(\hat{\mathbf{t}}) = \left\| \hat{\mathbf{t}} - \mathbf{t}_g \right\|_2. \tag{18}$$

### 4.2. Model Evaluation Experiment

Following PointNetLK [5], we train and evaluate PointCNT on ModelNet40. During the training, the rigid transformation $T_g$ is randomly generated, where the rotation is in the range of [0, 45] degrees with arbitrarily chosen axes, and translation is in the range [0, 0.8]. For a fair comparison, initial translations for testing are in the range [0, 0.3], and initial rotations are in the range of [0, 80] degrees. The traditional methods, ICP and one-stage methods, PointNetLK, PCRNet and FMR, and two-stage methods, DCP, GeoTransformer and PointDSC, are selected as the baseline.

#### 4.2.1. Train and Test on Same Object Categories

We use 20 ModelNet40 object categories to train our model and use the same 20 object categories to test our model. The results are shown in Figure 9 and Table 3. When the initial rotation angle is less than 40 degrees, the RRE, RTE and RR of PointCNT are close to those of the two-stage model. When the initial rotation angle is greater than 40 degrees, the RR of PointCNT is slightly lower than that of GeoTransformer and PointDSC, but it still exceeds DCP, traditional methods and one-stage methods. Compared with traditional methods and other one-stage methods, PointDSC is less sensitive to the initial position of the point cloud. This is because we utilize complex network theory to design a feature extraction network ComKP-CNN, which can extract the point cloud features better, and we explicitly embed the coordinate information and distance information to features in the feature embedding module. The results also show that compared with the traditional methods and other one-stage methods, our method is insensitive to the initial angle and achieves registration accuracy similar to that of two-stage methods.

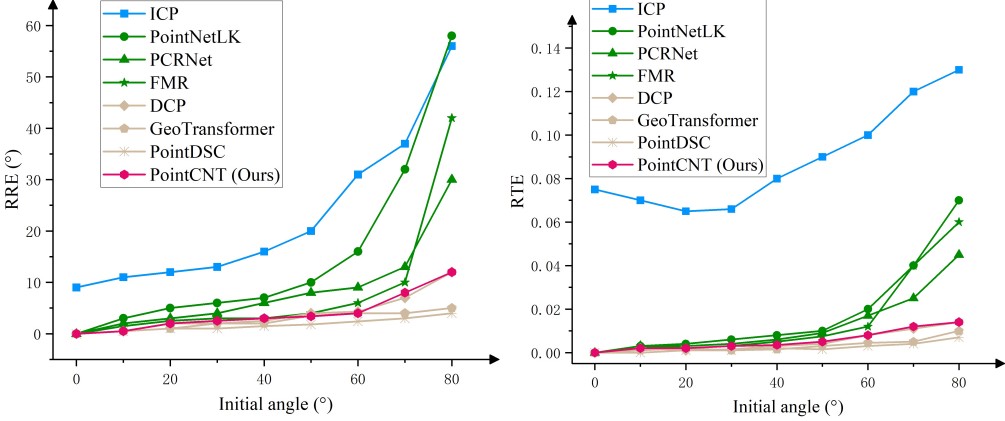

**Figure 9.** Comparison results of different methods under the same categories.

**Table 3.** Comparison results of different methods under the same categories.

| | Initial Angle (°) | ICP | PointNetLK | PCRNet | FMR | DCP | GeoTransformer | PointDSC | PointCNT (Ours) |
|---|---|---|---|---|---|---|---|---|---|
| RRE | 0 | 9.0583 | 0.0291 | 0.0867 | 0.0559 | 0.0484 | 0.0341 | 0.0574 | 0.0081 |
| | 10 | 11.1343 | 3.1020 | 2.1663 | 1.5260 | 0.6368 | 0.6518 | 0.5333 | 0.5488 |
| | 20 | 12.0865 | 5.1491 | 3.1363 | 2.6720 | 2.1669 | 1.1784 | 1.0502 | 2.1426 |
| | 30 | 13.1300 | 6.0747 | 4.0494 | 3.1946 | 2.0129 | 2.0468 | 1.1013 | 2.5567 |
| | 40 | 16.0468 | 7.0898 | 6.0083 | 3.1537 | 2.6155 | 2.0655 | 1.5227 | 3.1776 |
| | 50 | 20.1213 | 10.0190 | 8.0598 | 4.0932 | 4.1018 | 3.6256 | 1.8021 | 3.4471 |
| | 60 | 31.1141 | 16.1352 | 9.0943 | 6.0841 | 4.3775 | 4.0737 | 2.5845 | 4.0157 |
| | 70 | 37.1181 | 32.1661 | 13.0590 | 10.0162 | 7.1807 | 4.0254 | 3.0133 | 8.1355 |
| | 80 | 56.0147 | 58.1466 | 30.1745 | 42.0883 | 12.1429 | 5.1880 | 4.0809 | 12.1752 |
| RTE | 0 | 0.0752 | 0.0001 | 0.0001 | 0.0003 | 0.0003 | 0.0001 | 0.0002 | 0.0002 |
| | 10 | 0.0702 | 0.0031 | 0.0031 | 0.0031 | 0.0020 | 0.0011 | 0.0002 | 0.0021 |
| | 20 | 0.0653 | 0.0041 | 0.0031 | 0.0021 | 0.0011 | 0.0020 | 0.0012 | 0.0021 |
| | 30 | 0.0661 | 0.0062 | 0.0041 | 0.0031 | 0.0018 | 0.0012 | 0.0010 | 0.0031 |
| | 40 | 0.0801 | 0.0080 | 0.0062 | 0.0050 | 0.0033 | 0.0018 | 0.0020 | 0.0037 |
| | 50 | 0.0903 | 0.0102 | 0.0090 | 0.0075 | 0.0043 | 0.0031 | 0.0017 | 0.0053 |
| | 60 | 0.1000 | 0.0202 | 0.0171 | 0.0121 | 0.0083 | 0.0047 | 0.0033 | 0.0083 |
| | 70 | 0.1201 | 0.0402 | 0.0251 | 0.0402 | 0.0111 | 0.0051 | 0.0042 | 0.0122 |
| | 80 | 0.1300 | 0.0701 | 0.0452 | 0.0601 | 0.0141 | 0.0102 | 0.0072 | 0.0140 |

### 4.2.2. Train and Test on Different Object Categories

We train PointCNT with 20 ModelNet40 object categories and then test PointCNT with another 20 object categories. The results are shown in Figure 10 and Table 4. The performance of our model is obviously better than that of other one-stage methods, which shows that our model has good generalization performance.

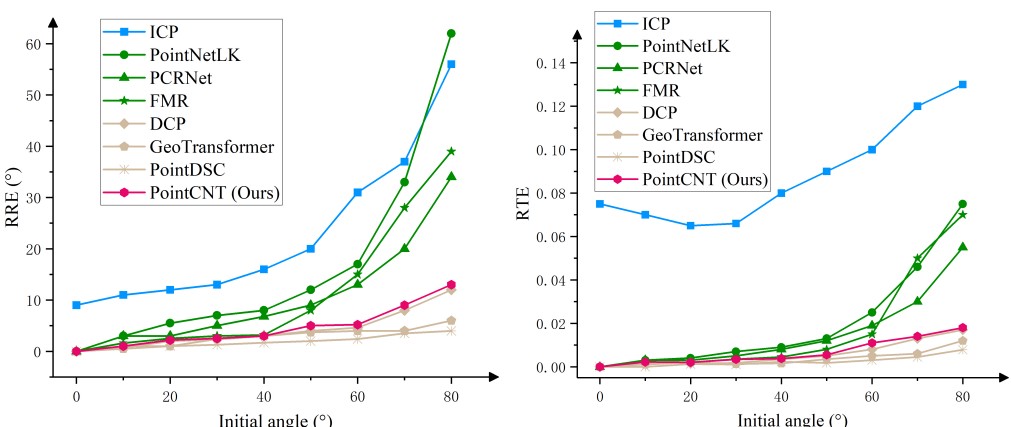

**Figure 10.** Comparison results of different methods under the different categories.

**Table 4.** Comparison results of different methods under the same categories.

| | Initial Angle (°) | ICP | PointNetLK | PCRNet | FMR | DCP | GeoTransformer | PointDSC | PointCNT (Ours) |
|---|---|---|---|---|---|---|---|---|---|
| RRE | 0 | 9.0959 | 0.1845 | 0.1469 | 0.0092 | 0.0133 | 0.0554 | 0.1145 | 0.0278 |
| | 10 | 11.0771 | 3.1107 | 3.1902 | 1.6782 | 0.7943 | 0.5084 | 1.1477 | 1.0007 |
| | 20 | 12.0457 | 5.6493 | 3.0634 | 2.6482 | 2.1269 | 1.1253 | 1.1978 | 2.3915 |
| | 30 | 13.0583 | 7.0153 | 5.1618 | 3.0255 | 2.3756 | 2.4167 | 1.3391 | 2.6777 |
| | 40 | 16.0784 | 8.1973 | 6.9155 | 3.2291 | 3.1534 | 3.0080 | 1.7222 | 3.0906 |
| | 50 | 20.0064 | 12.1962 | 9.0828 | 8.1173 | 4.0734 | 3.8179 | 2.1880 | 5.1831 |
| | 60 | 31.1160 | 17.0211 | 13.1846 | 15.0505 | 4.6581 | 4.0699 | 2.5154 | 5.3267 |
| | 70 | 37.1035 | 33.1767 | 20.1443 | 28.1324 | 8.1459 | 4.0913 | 3.5691 | 9.0556 |
| | 80 | 56.0563 | 62.0186 | 34.0484 | 39.1745 | 12.1727 | 6.1058 | 4.1685 | 13.0916 |

**Table 4.** *Cont.*

| | Initial Angle (°) | ICP | PointNetLK | PCRNet | FMR | DCP | GeoTransformer | PointDSC | PointCNT (Ours) |
|---|---|---|---|---|---|---|---|---|---|
| | 0 | 0.0749 | 0.0001 | 0.0000 | 0.0000 | 0.0000 | 0.0000 | 0.0000 | 0.0000 |
| | 10 | 0.0699 | 0.0030 | 0.0029 | 0.0020 | 0.0020 | 0.0010 | 0.0000 | 0.0022 |
| | 20 | 0.0650 | 0.0041 | 0.0029 | 0.0019 | 0.0014 | 0.0021 | 0.0011 | 0.0020 |
| | 30 | 0.0659 | 0.0070 | 0.0050 | 0.0036 | 0.0018 | 0.0014 | 0.0009 | 0.0033 |
| RTE | 40 | 0.0799 | 0.0090 | 0.0080 | 0.0044 | 0.0034 | 0.0016 | 0.0024 | 0.0037 |
| | 50 | 0.0901 | 0.0130 | 0.0119 | 0.0079 | 0.0050 | 0.0036 | 0.0017 | 0.0055 |
| | 60 | 0.1185 | 0.0251 | 0.0190 | 0.0150 | 0.0079 | 0.0050 | 0.0031 | 0.0110 |
| | 70 | 0.1150 | 0.0459 | 0.0300 | 0.0501 | 0.0130 | 0.0060 | 0.0045 | 0.0140 |
| | 80 | 0.1271 | 0.0750 | 0.0549 | 0.0701 | 0.0169 | 0.0120 | 0.0079 | 0.0181 |

### 4.2.3. Gaussian Noise Experiments

We conduct experiments to study the robustness of PointCNT to noise. PointCNT is trained and tested on the same 20 object categories of ModelNet40. The range of the standard deviation of Gaussian noise is [0, 0.05]. The results are shown in Figure 11 and Table 5. Our model is robust to noise, and the registration results are almost unaffected by noise. This is because GBSelf-attention and FBSelf-attention are nonlocal neural networks that can perceive global point features.

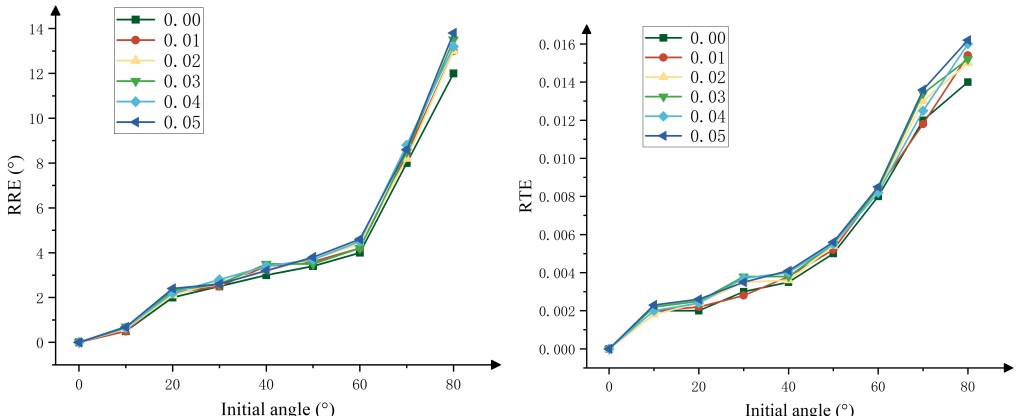

**Figure 11.** Comparison results of different Gaussian noises.

**Table 5.** Comparison results of different methods under the same categories.

| | Gaussian Standard Deviation | 0 | 0.01 | 0.02 | 0.03 | 0.04 | 0.05 |
|---|---|---|---|---|---|---|---|
| | 0 | 0.0029 | 0.0123 | 0.0133 | 0.0125 | 0.0071 | 0.0002 |
| | 10 | 0.5095 | 0.4869 | 0.5851 | 0.7079 | 0.6462 | 0.6709 |
| | 20 | 2.0118 | 2.1917 | 2.1138 | 2.3090 | 2.2119 | 2.3873 |
| | 30 | 2.5086 | 2.5101 | 2.6973 | 2.6055 | 2.7947 | 2.6089 |
| RRE | 40 | 3.0057 | 3.4008 | 3.1896 | 3.5051 | 3.4001 | 3.1924 |
| | 50 | 3.3896 | 3.6013 | 3.6986 | 3.4945 | 3.7084 | 3.7896 |
| | 60 | 3.9989 | 4.2122 | 4.3876 | 4.1903 | 4.4972 | 4.5954 |
| | 70 | 7.9855 | 8.3915 | 8.1942 | 8.4931 | 8.7858 | 8.6007 |
| | 80 | 11.9947 | 13.0138 | 13.0043 | 13.4954 | 13.1888 | 13.7911 |
| | 0 | 0.0000 | 0.0001 | 0.0000 | 0.0001 | 0.0000 | 0.0001 |
| | 10 | 0.0020 | 0.0020 | 0.0017 | 0.0021 | 0.0021 | 0.0024 |
| | 20 | 0.0019 | 0.0022 | 0.0025 | 0.0024 | 0.0024 | 0.0026 |
| | 30 | 0.0029 | 0.0029 | 0.0035 | 0.0038 | 0.0036 | 0.0036 |
| RTE | 40 | 0.0036 | 0.0039 | 0.0036 | 0.0037 | 0.0040 | 0.0042 |
| | 50 | 0.0051 | 0.0053 | 0.0054 | 0.0054 | 0.0055 | 0.0056 |
| | 60 | 0.0082 | 0.0084 | 0.0083 | 0.0083 | 0.0083 | 0.0084 |
| | 70 | 0.0119 | 0.0118 | 0.0130 | 0.0133 | 0.0125 | 0.0135 |
| | 80 | 0.0143 | 0.0155 | 0.0150 | 0.0152 | 0.0161 | 0.0161 |

#### 4.2.4. Partial Overlap Experiments

Partial overlap is a problem that point cloud registration has to face. A model has practical application value only if it can achieve acceptable registration results in the case of partial overlap. We manually remove part of the point cloud to compare the performance on the partial overlap. PointCNT is trained and tested on the same 20 object categories of ModelNet40. The range of the standard deviation of Gaussian noise is 0.05. The results are shown in Table 6, and the qualitative visualization results are shown in Figure 12. PointCNT achieves a registration result similar to two-stage methods, and the computation speed is faster than that of two-stage methods. This is because our model is end to end and does not need to find correspondences. The RR of our model is much higher and the RRE and RTE are much lower than those of traditional methods and other one-stage and two-stage methods. This is because FBSelf-attention enables the source point cloud and the template point cloud to perceive each other's features so that the geometric consistency of the overlapping part can be transmitted interactively between the two point clouds.

**Table 6.** Comparison results of different methods in the case of partial overlap.

| Model | RRE (°) | RTE | RR (%) | Time (s) |
|---|---|---|---|---|
| ICP | 17.3752 | 0.0253 | 82.3 | 0.12 |
| PointNetLK | 17.3752 | 0.0253 | 82.3 | 0.12 |
| PCRNet | 9.5863 | 0.0229 | 85.7 | 0.16 |
| FMR | 8.8724 | 0.0183 | 88.2 | 0.08 |
| DCP | 4.7283 | 0.0067 | 95.2 | 0.21 |
| GeoTransformer | 3.6878 | 0.0042 | 97.1 | 0.23 |
| PointDSC | 3.4586 | 0.0036 | 97.3 | 0.24 |
| PointCNT (Ours) | 4.5128 | 0.0064 | 96.4 | 0.15 |

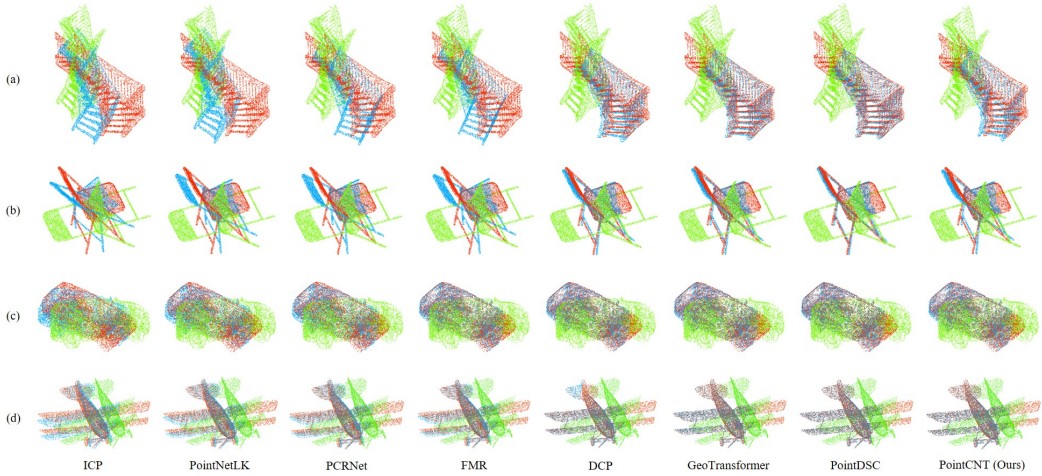

**Figure 12.** Partial overlap qualitative visualization registration results of different methods at different initial angles. The initial angles of (**a**–**d**) are 80, 60, 40 and 20, respectively. Green represents the source point cloud, red represents the template point cloud, and blue represents the registered source point cloud.

#### 4.2.5. Effectiveness of PointCNT

In order to verify the effectiveness of the proposed model on different types of datasets, we carried out experiments on the indoor dataset 3DMatch and the outdoor dataset KITTI. We use 3DMatch training data preprocessed by [53]. We split KITTI into two groups, training and testing. The training group includes 00–07 sequences, and the testing group includes 08–10 sequences. As shown in Table 7, PointCNT achieves good point cloud registration results on 3DMatch and KITTI, which proves the effectiveness of our model. Kitti is a natural object dataset. The excellent performance of the model on KITTI proves

that the feature extraction capability will not be limited according to the attributes and characteristics of the point cloud spatial distribution.

**Table 7.** Point cloud registration results of PointCNT on 3DMatch and KITTI.

| Dataset | RRE (°) | RTE (cm) | RR (%) |
|---|---|---|---|
| 3DMatch | 0.3258 | 7.3854 | 93.6 |
| KITTI | 0.2614 | 7.9368 | 97.2 |

### 4.3. Ablation Experiments

We conduct ablation experiments on ModelNet40 to study the effects of ComKP-CNN, self-supervised module, coordinate embedding, distance embedding, FBCross-attention, max pooling as the symmetric function and rotation quaternion as the point cloud rotation representation on the model. The effect of each module on registration accuracy improvement is verified. PointCNT is trained and tested on the same 20 object categories of ModelNet40 with partial overlap. The range of the standard deviation of Gaussian noise is 0.05. The results are shown in Table 8.

**Table 8.** The results of ablation experiments.

| | CK | SS | CE | DE | FB | MP | AP | RQ | RM | RRE (°) | RTE | RR (%) |
|---|---|---|---|---|---|---|---|---|---|---|---|---|
| baseline | | | | | | | ✓ | | ✓ | 6.7322 | 0.0135 | 85.9 |
| CK | ✓ | | | | | | ✓ | | ✓ | 5.3264 | 0.0083 | 89.3 |
| SS | | ✓ | | | | | ✓ | | ✓ | 5.7217 | 0.0087 | 87.4 |
| CE | | | ✓ | | | | ✓ | | ✓ | 5.6429 | 0.0086 | 87.6 |
| DE | | | | ✓ | | | ✓ | | ✓ | 5.8141 | 0.0088 | 86.9 |
| FB | | | | | ✓ | | ✓ | | ✓ | 5.5833 | 0.0085 | 88.1 |
| MP | | | | | | ✓ | | | ✓ | 6.0135 | 0.0090 | 86.6 |
| RQ | | | | | | | ✓ | ✓ | | 6.1078 | 0.0091 | 86.4 |
| SS+CE+DE+FB+MP+RQ | | ✓ | ✓ | ✓ | ✓ | ✓ | | ✓ | | 4.8463 | 0.0075 | 94.1 |
| CK+CE+DE+FB+MP+RQ | ✓ | | ✓ | ✓ | ✓ | ✓ | | ✓ | | 4.7234 | 0.0067 | 95.2 |
| CK+SS+DE+FB+MP+RQ | ✓ | ✓ | | ✓ | ✓ | ✓ | | ✓ | | 4.7832 | 0.0070 | 94.8 |
| CK+SS+CE+FB+MP+RQ | ✓ | ✓ | ✓ | | ✓ | ✓ | | ✓ | | 4.7138 | 0.0066 | 95.3 |
| CK+SS+CE+DE+MP+RQ | ✓ | ✓ | ✓ | ✓ | | ✓ | | ✓ | | 4.8195 | 0.0073 | 94.4 |
| CK+SS+CE+DE+FB+AP+RQ | ✓ | ✓ | ✓ | ✓ | ✓ | | ✓ | ✓ | | 4.6618 | 0.0069 | 95.6 |
| CK+SS+CE+DE+FB+MP+RM | ✓ | ✓ | ✓ | ✓ | ✓ | ✓ | | | ✓ | 4.6576 | 0.0068 | 95.8 |
| CK+SS+CE+DE+FB+MP+RQ (Ours) | ✓ | ✓ | ✓ | ✓ | ✓ | ✓ | | ✓ | | 4.5128 | 0.0064 | 96.4 |

Note: CK is ComKP-CNN, SS is self-supervised module, CE is coordinate embedding, DE is distance embedding, FB is FBCross-attention, MP is max pooling, AP is average pooling, RQ is rotation quaternion and RM is rotation matrix.

The results show that the complex ComKP-CNN, self-supervised module, coordinate embedding, distance embedding and FBCross-attention all improve the model registration accuracy. Among them, ComKP-CNN contributes the most to the model and reduces RRE and RTE by 0.3335 and 0.0011, respectively, and RR increases by 2.3%, which indicates that ComKP-CNN designed in this paper is effective for extracting the point cloud features. The registration accuracy improvement by FBCross-attention is only second to that of ComKP-CNN. This is because FBCross-attention realizes the interactive propagation of point cloud features, including geometric consistency and coordinate difference between source and template. Table 8 shows that the registration effect when max pooling is used as the symmetric function in the model is better than that when average pooling is used as the symmetric function. This is because average pooling is too smooth, which makes the difference in global features of different point clouds become insignificant, while max pooling does not have such a problem. When the rotation quaternion is used to represent the rotation of the point cloud rather than the rotation matrix, the model registration effect is better, which is the same as the experimental result of 3DRegNet [29].

Experiments are carried out to study the influence of our designed feature extraction module (ComKP-CNN and self-supervised module), feature embedding module (coordinate embedding and distance embedding) and feature fusion module (FBCross-attention

and symmetry function) on the registration results under different initial angles. PointCNT is trained and tested on the same 20 object categories of ModelNet40 with partial overlap. The range of standard deviation of Gaussian noise is 0.05. The results are shown in Table 9.

**Table 9.** Comparison results of different models with and without ComKP-CNN.

| Initial Angle | 20° | | 40° | | 60° | | 80° | |
|---|---|---|---|---|---|---|---|---|
| Metrics | RRE (°) | RTE | RRE (°) | RTE | RRE (°) | RTE | RRE (°) | RTE |
| Using ComKP-CNN | 2.4273 | 0.0026 | 3.2351 | 0.0041 | 4.6168 | 0.0085 | 13.8413 | 0.0162 |
| Using KP-CNN | 2.5185 | 0.0031 | 3.5017 | 0.0048 | 4.8912 | 0.0097 | 14.376 | 0.0177 |
| Using self-supervised modeule | 2.4273 | 0.0026 | 3.2351 | 0.0041 | 4.6168 | 0.0085 | 13.8413 | 0.0162 |
| No self-supervised modeule | 2.5032 | 0.0029 | 3.4926 | 0.0045 | 4.8128 | 0.0092 | 14.1734 | 0.0169 |
| Using coordinate embedding | 2.4273 | 0.0026 | 3.2351 | 0.0041 | 4.6168 | 0.0085 | 13.8413 | 0.0162 |
| No coordinate embedding | 2.4984 | 0.0028 | 3.4586 | 0.0044 | 4.8326 | 0.0091 | 14.0128 | 0.0168 |
| Using distance embedding | 2.4273 | 0.0026 | 3.2351 | 0.0041 | 4.6168 | 0.0085 | 13.8413 | 0.0162 |
| No distance embedding | 2.4815 | 0.0029 | 3.4125 | 0.0044 | 4.8402 | 0.0091 | 13.9821 | 0.0166 |
| Using FBCross-attention | 2.4273 | 0.0026 | 3.2351 | 0.0041 | 4.6168 | 0.0085 | 13.8413 | 0.0162 |
| No FBCross-attention | 2.5148 | 0.003 | 3.4824 | 0.0046 | 4.8621 | 0.0094 | 14.2675 | 0.0172 |
| Using max pooling | 2.4273 | 0.0026 | 3.2351 | 0.0041 | 4.6168 | 0.0085 | 13.8413 | 0.0162 |
| Using average pooling | 2.4637 | 0.0028 | 3.3861 | 0.0043 | 4.8236 | 0.0087 | 13.9643 | 0.0165 |

The results show that our designed feature extraction module, feature embedding module and feature fusion module can improve the registration accuracy of the model when the point cloud has different initial angles. And the larger the initial angle, the more obvious the effect of the module we designed on improving the accuracy of registration. Experiments prove the effectiveness of the designed feature extraction module, feature embedding module and feature fusion module in different initial angles.

### 4.4. Effectiveness of ComKP-CNN

One of the important contributions of this paper is to propose a DNN design method and design a new point cloud feature extraction framework ComKP-CNN. Therefore, we use ComKP-CNN to replace the feature extraction module of other point cloud registration frameworks to verify the effectiveness of ComKP-CNN. Table 10 shows that ComKP-CNN reduces the registration errors of PointNetLK, PCRNet, FMR, DCP, PointDSC and GeoTransformer to varying degrees and improves the registration accuracy. This indicates the correctness of our DNN design idea based on complex network theory. This idea is expected to be extended to the design of deep learning frameworks in other fields.

**Table 10.** Comparison results of different models with and without ComKP-CNN.

| | | PointNetLK | PCRNet | FMR | DCP | GeoTransformer | PointDSC |
|---|---|---|---|---|---|---|---|
| Without ComKP-CNN | RRE (°) | 17.3752 | 9.5863 | 8.8724 | 4.7283 | 3.6878 | 3.4586 |
| | RTE | 0.0253 | 0.0229 | 0.0183 | 0.0067 | 0.0042 | 0.0036 |
| | RR (%) | 82.3 | 85.7 | 88.2 | 95.2 | 97.1 | 97.3 |
| With ComKP-CNN | RRE (°) | 14.8463 | 7.8362 | 7.2156 | 3.6748 | 3.1163 | 3.0376 |
| | RTE | 0.0221 | 0.0204 | 0.0168 | 0.0055 | 0.0037 | 0.0034 |
| | RR (%) | 86.6 | 87.8 | 90.4 | 96.3 | 97.7 | 97.9 |

## 5. Discussion

It can be concluded that PointCNT is a novel and competitive registration algorithm for partial assignment tasks from the above extensive experiments. Mainly, some meaningful discussions are summarized below.

We conducted an experiment on the same object categories and different object categories on ModelNet40. The experiment shows that the accuracy of deep learning methods is significantly better than traditional methods, and our method's registration accuracy is close to that of two-stage methods, and it has good generalization performance. The noise

experiment shows that our method is minimally affected by noise. Our method shows advanced robustness in point cloud registration under noise interference. Partial overlap is a problem that point cloud registration has to face. The partial overlay experiment shows that our method achieves high registration accuracy in cases where point clouds only partially overlap, and the registration accuracy is much higher than other one-stage methods. The ablation experiment shows that ComKP-CNN contributes the most to the model and reduces RRE and RTE by 0.3335 and 0.0011, respectively, and RR increases by 2.3%, which proves the effectiveness of the deep learning model design method based on complex network theory proposed in this paper.

However, the point cloud registration accuracy of the proposed method is still lower than that of the two-stage method. In addition, although the method proposed in this paper is a one-stage method with fast inference speed and real-time performance, PointCNT is still complex and not easy to deploy.

## 6. Conclusions

We propose an efficient and high-precision one-stage point cloud registration method. The DNN design method based on complex network theory can not only be used for the design of point cloud feature extraction network but is also expected to be applied to the design of feature extraction networks for image, text, voice and other types of data. The results show that the designed ComKP-CNN can efficiently extract the features of point clouds, significantly reduce the error of point cloud registration and is expected to be applied to 3D target detection, semantic segmentation and other tasks. The results also show that the self-monitoring module is helpful for the model to better extract the features of the point cloud. In addition, the feature embedding module explicitly embeds the geometric information into the point cloud feature, which is helpful for point cloud registration. We also find that FBCross-attention makes the features of source point cloud and template point cloud perceptible to each other and improves the point cloud registration accuracy.

The proposed method of explaining and designing a deep learning model based on complex network theory is a novel idea. The method proposed in this paper is expected to be applied to 3D reconstruction, map reconstruction, digital twinning and other fields. In the future, we will carry out further detailed research on this method in the field of image recognition. We will also carry out research on model compression and model deployment.

**Author Contributions:** X.W. (Xin Wu), X.W. (Xiaolong Wei) and H.X. were responsible for the overall algorithm design and experimental design and wrote the paper. C.L., Y.H. and Y.Y. were responsible for the coding and experimental execution. W.H. was responsible for correcting complex papers. All authors have read and agreed to the published version of the manuscript.

**Funding:** This research was funded by the National Natural Science Foundation of China grant number 12075319, the National Natural Science Foundation of China grant number 11805277 and the National Natural Science Foundation of China grant number 51975583.

**Data Availability Statement:** Data available in a publicly accessible repository.

**Acknowledgments:** The public dataset used in this article is ModelNet40, 3DMatch and KITTI.

**Conflicts of Interest:** The authors declare no conflict of interest.

## Abbreviations

The following abbreviations are used in this manuscript:

| | |
|---|---|
| 2D | Two-Dimensional |
| 3D | Three-Dimensional |
| 3DRegNet | Deep Neural Network for 3D Point Registration |
| 3Dsc | 3D Shape Context |
| 4PCS | 4-Points Congruent Sets |
| ANNs | Artificial Neural Networks |
| CAD | Computer-Aided Design |

| ComKP-CNN | Complex Kernel Point Convolution Neural Network |
|---|---|
| ConvBlock | KPConv Block |
| DCP | Deep Closest Point |
| DeepVCP | Deep Virtual Corresponding Points |
| DGR | Deep Global Registration |
| FBCross-attention | Feature-based Cross-attention |
| FMR | Feature-Metric Registration |
| FPFH | Fast Point Feature Histogram |
| GBSelf-attention | Geometric-based Self-attention |
| GeoTransformer | Geometric Transformer |
| Go-ICP | A Globally Optimal Solution to 3D ICP Point-set Registration |
| GPU | Graphic Processing Unit |
| ICP | Iterative Closest Point |
| KP-CNN | Kernel Point Convolution Neural Network |
| KPConv | Kernel Point Convolution |
| LiDAR | Light Detection and Ranging |
| LK | Lucas–Kanade Algorithm |
| MLP | Multilayer Perceptron |
| NDT | Normal Distributions Transform |
| PCA | Principal Component Analysis |
| PCRNet | Point Cloud Registration Network using PointNet Encoding |
| PFH | Point Feature Histogram |
| PointCNT | A One-Stage Point Cloud Registration Approach Based on Complex Network Theory |
| PointDSC | Robust Point Cloud Registration using Deep Spatial Consistency |
| PointNetLK | Robust and Efficient Point Cloud Registration using PointNet |
| PRNet | Partial Registration Network |
| RANSAC | Random Sample Consensus |
| ResBlock | Residual Blocks |
| RGM | Robust Point Cloud Registration Framework Based on Deep Graph Matching |
| RR | Registration Recall |
| RRE | Relative Rotation Error |
| RTE | Relative Translation Error |
| SVD | Singular Value Decomposition |

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
