# Peer review of "PointCNT: A One-Stage Point Cloud Registration Approach Based on Complex Network Theory"

_remotesensing, doi:10.3390/rs15143545_

Round 1

Reviewer 1 Report

1. This study proposes a point cloud multi-path feature extraction network method based on complex network theory for high-precision point cloud registration. In the research design, feature perception and fusion are mainly based on coordinate and distance information, and good registration accuracy is obtained. The structure of the article is complete and the verification process is detailed. It is a good research paper.

2. ComKP-CNN uses global features to estimate the rigid transformation matrix, which is different from the two-stage operation of finding the corresponding relationship and estimating the transformation matrix. This type of algorithm does not need to search for the corresponding relationship, so it is understandable that the calculation speed is faster, but whether the feature extraction capability will be limited according to the attributes and characteristics of the point cloud spatial distribution, this part of the suggestion can be supplemented. For example, the identification objects in this study are all man-made objects. If the objects are natural objects such as terrain or trees, these objects are less consistent in shape. Is it possible to use a non-two-stage algorithm and a local optimal solution to cause a large discrepancy? Quasi error?

3. The use of global distance calculation can effectively reduce the commission error in the overlapping parts of point clouds, but how to avoid including misclassified features in the calculation part of the feature fusion module, it is recommended to add another paragraph of text to explain.

4. Does the point cloud data used in this study only have three-dimensional coordinates and does not contain any other attributes? Considering that the latest LiDAR scanning instruments all have point cloud coloring functions. If the point cloud data contains more attributes, can it also be used as data of another dimension and included in the feature embedding module?

5. The sensitivity detection of the rotation angle of the test object to PRE and RTE is very strong evidence, which proves that the author's model does have good generalization performance and stability. However, whether the feature mutual perception performance of the source point clouds and the temporary point clouds can maintain a very good effect under various circumstances requires a more complete explanation.

6. It is recommended that the training and testing time required for the various algorithms included in this research be presented in a table and compared.

Reviewer 2 Report

1. Each acronym should be explained the first time it appears in the text, even if it appeared in the abstract. Check all abbreviations in text: each word should start with capital to explain an abbreviation.

2. Suggest  the authors to provide a comparative analysis of the various data sets as used in the literature survey preferably in a table to justify the outcome as claimed to be very efficient, as it is fuzzy.

3.  The  article should focus on a short paragraph to introduce what the rest of the paper contents will follow at the end of the Introduction section is missing. This paragraph is important; as it can enable the readers to understand what the following content will be and arouse their interest to continue reading the paper.

4. Section 3 – not clear in par with section 4 – fuzzy need clarity here.

5. Redraft all the titles and contents appropriately – Figure titles to be more precise rather than a paragraph at different sections.

6. The experimental results can be more organized to validate the theoretical ideas – Need a clarity in this section also.

7. Redraft the conclusion addressing the result findings appropriately. Also, include at least one future scope to it- which is more generalized.

as per the article contents there exists revision in English grammar and word choice as used at all sections.

Reviewer 3 Report

This paper deals with an exciting topic. The article has been read carefully, and some minor issues have been highlighted in order to be considered by the author(s).

#1 What is the motivation of this paper?

#2 What is the contribution and novelty of this paper?

#3 What is the advantage of this survey paper?

#4 Which evaluation metrics did you used for comparison?

#5 If possible, it would be good if image security domains would be reflected in the related work such as Adversarial image perturbations with distortions weighted by color on deep neural networks, Dual-Mode Method for Generating Adversarial Examples to Attack Deep Neural Networks, Toward Backdoor Attacks for Image Captioning Model in Deep Neural Networks, Advguard: fortifying deep neural networks against optimized adversarial example attack, Multi-model selective backdoor attack with different trigger positions.

#6 Author can clearly explain the challenges faced in the existing system and the motivation of the proposed system. 

#7 Meaning of the symbols used can be explained clearly. 

#8 What is the running time of proposed method. 

#9 The complexity analysis of the proposed method can be mentioned. 

#10 The limitation of the proposed work can be discussed. 

Round 2

Reviewer 2 Report

11.  The abstract must have precise quantification. It reads vaguely about efficiencies over prior works. The authors must put the benchmarks carefully and quantify the performance needs a minor revision here.

22. Each acronym should be explained the first time it appears in the text, even if it appeared in the abstract. Check all abbreviations in text: each word should start with capital to explain an abbreviation.

33. suggest the authors to provide a comparative analysis of the various algorithms as used in the literature survey preferably in a table to justify the outcome as claimed to be very efficient, where there is a small confusion.

43.   Section 3, Section 4  à need a clarity here, looks fuzzy.

84. Redraft all the titles and contents appropriately.

95. The experimental results can be more organized to validate the theoretical ideas, ideal to have the comparisons made in a table  – Need a clarity in this section which is missing.

16.  Redraft the conclusion addressing the result findings appropriately. Also, include at least one future scope to it as it is more generalized need in specific regarding the outcome of the proposed technique.

suggest the authors to recheck for the grammar and syntax of many sentences is poor and to correct them by more careful and thorough proof-reading.

Reviewer 3 Report

I recommend the acceptance.

Author Response

Thank you for your review.